# The Use of Esophageal Stents in the Management of Postoperative Fistulas—Current Status, Clinical Outcomes and Perspectives—Review

**DOI:** 10.3390/life13040966

**Published:** 2023-04-07

**Authors:** Cristian Gelu Rosianu, Petre Hoara, Florin Achim, Rodica Birla, Alexandra Bolocan, Ahmed Mohssen, Narcis Copca, Silviu Constantinoiu

**Affiliations:** 1Department of General Surgery, “Carol Davila” University of Medicine and Pharmacy, 050474 Bucharest, Romania; 2Gastroenterology Department, “Sfanta Maria” Clinical Hospital, 011172 Bucharest, Romania; 3Centre of Excelence in Esophageal Surgery, “Sfanta Maria” Clinical Hospital, 011172 Bucharest, Romania; 4Emergency University Hospital Bucharest, 050098 Bucharest, Romania; 5Second Department of Surgery, “Sfanta Maria” Clinical Hospital, 011172 Bucharest, Romania

**Keywords:** esophageal leak, endoscopic stent, endovacuum therapy, VACstent

## Abstract

Esophageal fistula remains one of the main postoperative complications, with the treatment often requiring the use of stents. This article reviews the updates on the use of endoscopic stents for the treatment of postoperative esophageal leakage in terms of indications, types of stents used, efficiency, specific complications and perspectives. Materials and Methods: We searched the PubMed and MEDLINE databases for the keywords postoperative esophageal anastomotic leak and postoperative esophageal anastomotic leak stent, and retrieved relevant papers published until December 2022. Results: The endoscopic discovery of the fistula is usually followed by the insertion of a fully covered esophageal stent. It has an efficiency of more than 60% in closing the fistula, and the failure is related to the delayed application of the method, a situation more suitable for endo vac therapy. The most common complication is migration, but life-threatening complications have also been described. The combination of the advantages of endoscopic stents and vacuum therapy is probably found in the emerging VACstent procedure. Conclusions: Although the competing approaches give promising results, this method has a well-defined place in the treatment of esophageal fistulas, and it is probably necessary to refine the indications for each individual procedure.

## 1. Introduction

Despite advances in minimally invasive surgical techniques, anastomotic fistula remains a serious complication, causing significant morbidity and mortality. The rate of anastomotic fistula after esophagectomy varies in the literature between 2 and 20%, being one of the most common surgical complications [1,2,3].

In the case of anastomotic fistulas, mortality rates of up to 25–45% have been reported [4]. Most often, fistulas appear 3–5 days after surgery and are associated with prolonged hospitalization in the intensive care unit [5]. Some studies show that neoadjuvant therapy does not influence the rate of anastomotic fistulas [6]. The mechanisms involved in the occurrence of anastomotic fistulas can be different, from the classic ones, i.e., inadequate perfusion, tissue trauma, local hematoma, tension in the suture and tumoral invasion, to the most specific ones, including stapler misfire or inappropriate use of the type of stapler for a specific tissue [7].

The key to managing a fistula and reducing its consequences includes early recognition and prompt initiation of treatment [8]. Anastomotic leakage results in longer hospital stays, worse long-term outcomes and decreased survival in postesophagectomy cancer patients [4]. Although primary repair of fistulas has been effective over time, it has nevertheless been associated with significant morbidity and mortality [9,10]. Initially, conservative management and surgical reintervention were the only treatment options but were often associated with poor outcomes (40–100% mortality in reoperative surgery and up to 40% morbidity and mortality for conservative management) [11,12].

Therefore, the introduction of endoscopic procedures for the management of these fistulas has become an attractive option. Depending on the size of the fistula, different endoscopic techniques are used, such as clips, fibrin injection, self-expanding metallic stents, drainage with trans anastomotic tubes or endoscopic vacuum therapy [5]. The choice of endoscopic technique often depends only on the expertise and preferences of the medical team on the availability of devices or the timing and size of the fistula. The studies of Schweeigert et al. and Nguyen et al. show that the endoscopic treatment of anastomotic fistulas led to a decrease in hospitalization, a reduction in the need for gastrointestinal diversion, better control of sepsis and a reduction in postoperative mortality compared to surgical treatment [4,13]. This review aims to present the current status of the treatment of esophageal fistulas by endoscopic stenting in terms of indications, types of stents used, the efficiency of the method, specific complications and its perspectives.

## 2. Methods (See Table 1 and Table 2)

The article is based on the analysis of data considered relevant for the chosen topic from the studies identified in PubMed Central (PMC) and MEDLINE Complete (EBSCO) since 2011, but also on the experience in endoscopic stenting of postoperative esophageal fistulas in the General and Esophageal Surgery Clinic of the Sfanta Maria Clinical Hospital, Bucharest, Romania. Trials were sought and used, as well as data from updates of studies, original articles or reviews, regarding esophageal stenting for postoperative leaks. For a sensitive search strategy, the terms used in search engines were “postoperative esophageal anastomotic leak” and “postoperative esophageal anastomotic leak and stent”. The article focused on updated data about the types of esophageal stent with characteristics, indication for use, associated procedures, efficiency and failure, contraindication, complication and alternative endoscopic treatment. Only these studies and papers were considered eligible, thus being taken into account in the elaboration of this article. Two authors (C.G.R. and R.B.) selected the articles considered relevant, preferring peer-reviewed articles from highly ranked journals written in English. The decision to select an item was made by agreement of the two. A number of 100 articles were identified for the period 2011–2023, which included the keywords used in the database search, ten reviews, six systematic reviews, four meta-analyses, three case reports, thirty-four observational studies and five comparative studies (Table 3). The reference list from each selected article was screened for additional relevant information. We excluded unpublished data from abstracts, contained in volumes from various congresses or conferences, as we excluded papers that were not in English.
life-13-00966-t001_Table 1Table 1The search strategy summary.ItemsSpecificationDatabases and other sources searchedPubMed Central (PMC), MEDLINE Complete (EBSCO)Search terms used (including MeSH and free text search terms and filters)Search strategy (see Table 2)Timeframe2011–2023Inclusion and exclusion criteria (study type, language restrictions etc.).Inclusion criteria: meta-analyses; trials studies; clinical trials and updates of clinical trials; reviews; original articles; only studies/papers/journals written in English. Exclusion criteria: unpublished data from abstracts contained in volumes from various congresses or conferences; papers that were not in English.Selection process (who conducted the selection, whether it was conducted independently, how consensus was obtained, etc.)C.G.R. performed the search in the databases, according to the presented criteria. If a study appears relevant by at least one reviewer—R.B. and C.G.R.—the full-text article was retrieved and checked. The selection of full-text articles was made by two reviewers independently: C.G.R. and R.B. Assessing content validity required subjective judgment from the reviewers. The citation number was an important selection criterion. Differences were discussed and if consensus could not be reached between the two reviewers, we requested the consultation and recommendation of a third reviewer (S.C.). The reference list from each selected article was screened for additionalrelevant information
life-13-00966-t002_Table 2Table 2Systematic literature search for stenting of anastomotic leaks.Search TermsNumber of Articles1. Search: Postoperative anastomotic leak stent4132.Search: Postoperative esophageal anastomotic leak stent1383.Search: Postoperative esophageal anastomotic leak stent; Filters: Full text, English1234.Search: Postoperative esophageal anastomotic leak stent; Filters: Full text, English, from 2011 to 2023100Final number after review for inclusion and exclusion criteria and addition of articles from references ^a^62^a^ Exclusion criteria: No abstract available; non-postoperative esophageal leak; anastomotic leak not a major focus of article; not published in English; case reports without an esophageal stent.


## 3. Indication for Esophageal Stenting

Apart from the indications for fistulas, documented by imaging, Speer et al. reported the use of this method for prophylactic purposes, the stents being placed in the anastomosis “at risk”, indicated by the attending surgeon in the first 24 postoperative hours [14]. Most studies emphasize the fact that the stents were inserted at the time of the endoscopic detection of the fistula. Smith et al. showed that the indication for stenting was given by the presence of the fistula, regardless of the size of the anastomotic defect, the extent of the associated mediastinal contamination or the moment of fistula identification [15]. In the study by Bohle et al., the use of stents was according to the size of the fistula; in 82% of the patients in whom a stent was inserted, the size of the fistula was less than 1/3 of the anastomotic circumference and 18% presented the size of the fistula between 1/3 and 2/3 of esophageal circumference [16]. El-Sourani et al. in a 2020 study recommended the use of stents in small defects (<6 mm) without extraluminal cavity in stable, non-septic patients [17].

D’Cunha and colleagues [18] recommended only stenting if the fistula was diagnosed within 24 h of its occurrence with minimal mediastinal contamination, or instead, in the case of uncontrolled fistula, diagnosed after 24 h or when there was imaging evidence of mediastinal contamination, recommended the association of a drainage procedure.

Although specific recommendations vary among authors, there is a consensus that the use of stents must be selective and individualized, taking into account the particular anatomy of the patient and the etiology of the fistula, as well as the clinical status, the time until the diagnosis, the existence of an extraluminal cavity and the condition of the tissues adjacent to the anastomotic defect. There is also consensus that continuous reassessment of the patient’s condition is necessary to determine if the fistula has been sealed and if additional interventions are needed. An esophagogram within 24 h is recommended to confirm that the fistula is sealed [19].

## 4. Types of Esophageal Stents

In most studies, fully covered metal stents are used as a rule, and only sometimes are partially covered ones used. The size of the stent was selected at the time of endoscopy, based on the size of the anastomosis, and varied between 16 and 19 mm in diameter, the most frequently used being the 19 mm stent in a study by Smith et al. [15] or 20–28 mm reported by Bohle et al. [16].

A variety of materials have been studied for this purpose, including partially or fully covered plastic and metal stents. Typically, these stents are left in place for 4–6 weeks, after which they are removed, and the fistula area is evaluated endoscopically [53]. In an analysis comparing the clinical success rate of different types of stents, no statistical differences were found between the use of self-expanding plastic stents (84%), fully covered self-expanding metal stents (85%) and partially covered self-expanding metal stents (86%); however, plastic stents (actually less and less used) were kept in place for a significantly longer period and showed a higher rate of migration than partially covered metallic stents [20].

The same authors show that the healing is monitored endoscopically, and in the case of suspicion of migration, a revision of the procedure (repositioning or replacement) is necessary. Most patients require a single stent, but van Boeckel et al. state that up to half of the patients may require two or more consecutive stents to heal the fistula [20].

## 5. Stenting Associated Procedures

Endoscopic verification of the correct position of the stent, with adequate coverage of the fistulous orifice, and placement under endoscopic visualization of a nasogastric tube, positioned through the stent with the tip in the gastric conduit, are associated procedures. Some authors recommend enteral feeding through a previously placed jejunostomy [15].

Due to the relatively high incidence of post-stent fistulas, secondary to migration, in cervical stenting, suture or nasal anchoring is recommended; the endoluminal suture of the stent to the mucosa is also useful for other locations, and whenever possible, the association of a laparoscopic gastrostomy for gastric decompression can be proposed. In addition, due to the poor results in patients in whom the fistula does not seal after 24–48 h, the same authors recommend prompt intervention, either with another attempt of stent placement, associated with adjunctive procedures, to increase the probability of sealing or surgical diversion [19].

Other authors have also proposed different fixation methods as another method of reducing the rate of stent migration [54]. On the other hand, Speer et al. in a study in 2016 did not observe the benefits of fixation procedures in preventing stent migration [14]. Likewise, Singer and colleagues in a study elaborated on 214 esophageal stenting procedures, which used stent fixation procedures in 5% of cases; endoscopic clips (9%), endoscopic sutures (73%) and transnasal sutures (18%) showed that the rate of stent migration was not different between those with and without fixation (*p* > 0.05) [21].

## 6. The Efficiency of the Method in the Treatment of Fistula

Fistula healing is monitored by endoscopy. Although the groups of patients included in the studies vary, the rate of successful fistula closure by stenting varies between 63.5 and 100% [16,22,23,24]. Segura et al. in a 2022 study reported fistula healing using a single stent with the covered double-layer metal stent (Niti S™ DOUBLE™ Esophageal Metal Stent Model) in a percentage of 75% and 100% in terms of using the method [25], and Haruštiak et al. revealed an efficiency of 84% in closing the fistula by stenting in an average time of 55.7 ± 27.11 days/patient (mean of 39 ± 24.30 days/stent) [26].

Eizaguirre et al. noted that stenting is more effective in thoracic fistula locations [27]. Regarding the influence of the size of the defect on the success rate, there are different reports. Hoeppner et al. [10] did not find any difference in clinical success rates comparing fistulas ≤10 mm versus those >10 mm, while Kim et al. [28] reported better results after the use of clips, fibrin injection or insertion of stents in fistulas smaller than 2 cm. Bohle et al. showed in their study that neoadjuvant radiochemotherapy, squamous cell carcinoma histology and tumor location were factors associated with stent failure [16]. Esophageal stenting may not be successful in some cases if the defect is large or if the viability of the tissue around the defect is low, which is the usual situation in patients with neoadjuvant radiochemotherapy [55].

Berlth et al. in a comparative study of stenting therapy and vacuum therapy showed comparable results for the management of anastomotic fistulas in terms of overall fistula closure rate—85.7% for vacuum therapy and 72.4% for stent therapy (*p* > 0.05). The average hospitalization in the therapy compartment was 6 days for vacuum therapy and 9 days for those with stents (*p* > 0.05), and the average hospitalization for patients with vacuum therapy was 39 days and for patients with stent placement stents 37 days; (*p* > 0.05) [29]. Mennigen et al. in a 2015 study on 45 patients with fistulas, 30 treated by stenting and 15 by endovacuum therapy showed an efficiency of stenting of 63.5% compared to 93.3% of vacuum therapy [30]. A meta-analysis performed in 2018 by Rausa et al. showed that the closure rate of esophageal fistulas is significantly higher after vacuum therapy than with the use of stents (cumulative rate ratio 5.51CI 95% (2.11–14.88); *p* < 0.001). In addition, vacuum therapy has a shorter duration of treatment (combined mean difference—9.0 days), a lower rate of major complications (*p* < 0.05) and in-hospital mortality (*p* < 0.05) [31]. El-Sourani et al. showed in a study in 2020 that the use of vacuum therapy in defects >15 mm led to an efficiency of 92.3% in critical patients [17]. Tavares et al. and Tustumidid et al. performed a meta-analysis in 2021 that included 23 articles and 559 patients and showed a 16% difference in fistula closure efficiency (risk difference (RD): 0.16; 95% CI: 0.05–0.27) and a 10% difference regarding postprocedural mortality (RD: −0.10; 95% CI: −0.18 to −0.02) using vacuum therapy compared to endoscopic stenting [32].

A multicenter study by Hallit et al. that included 37/69 patients with post-esophagectomy fistulas, published in 2021, showed an efficiency of 95% in closing the fistula using internal drainage (double pigtail stents (Cook^®^ Solus or Boston Scientific^®^, Marlborough, MA, USA, Advanix 7 or 10 Fr 3–5 cm stents) or combined therapy with the use of over-the-scope clips in some cases (OVESCO^®^ over-the-scope clip 100.10) and 77% by stenting, the choice of method being given by the size of the defect. A large anastomotic defect (>2 cm or one-third of the anastomotic circumference) or any size anastomotic defect associated with areas of necrotic mucosa led to the placement of an esophageal stent, and smaller anastomotic defects led to the choice of internal drainage [33]. Studies comparing the efficiency of new therapies (endovacuum or double pigtail) and classic stenting suggest better results for the new methods. Lazzarin et al. studied five patients with esogastric fistula after bariatric surgery treated with double pigtail and showed 100% healing after 30–40 days [56].

## 7. The Persistence of the Leak after Stent Placement

The persistence of the fistula after the insertion of esophageal stents is a failure with dire consequences for the patient. In order to reduce the consequences of a persistent post-stenting fistula, it is necessary that it be recognized and managed properly. The reported incidence of fistula persistence varies but may be seen in approximately 10–38% of patients [24,26,34,35,36]. Stephens et al. [19] proposed in 2014 a classification of persistent post-stent fistula into five types based on radiographic evaluation with water-soluble contrast: Type 1—proximal; Type 2—retrograde distal; Type 3—through holes in the film covering the stent; Type 4—between stents; and Type 5—migrated stent. The proximal type occurs when stenting was performed in regions where it was difficult to achieve a proximal seal; replacing it with a larger stent or placing the second stent over the large diameter region is the most appropriate option. Type 2 is encountered in cases where the stent has been placed in a region with a larger distal diameter; placement of a decompression PEG, an additional stent over the distal region or replacement of the stent with a larger one may be helpful. Type 3 is associated with technical difficulties when placing the stent or possibly when the stent is placed on suction; changing the stent is beneficial. Type 4 appears due to the failure of the stent-in-stent procedure, best resolved by replacing the proximal stent with a larger one. Type 5 occurs especially in cervical stenting, without fixation or anchorage, or in stents placed in the mid-oesophagus; it is necessary to change the stent with a larger one or to use a sort of fixation or anchor the stent to the nasal septum. These authors presented a study of 23 patients with persistent post-stent fistulas, 65% of patients being resolved by using stents (12/23—additional stent, some larger ones), percutaneus endoscopic gastrostomy (PEG) (3/23), observation (2/23) or surgery (6/23); the cases were encountered in patients with a significant delay between diagnosis and appropriate intervention [19].

Repeated stenting is used if the patient is stable or presents a particularly high surgical risk. If the fistula is not sealed within 24–48 h, it is unlikely to seal; depending on the patient’s condition, either the stent is replaced, or the fistula is patched with an adjuvant muscle flap if necessary, and surgical intervention is justified. However, in patients who fail to seal the fistula after two attempts or become unstable, prompt surgical diversion should be considered because persistent esophageal fistula has a poor prognosis [19].

Persistence of the fistula after the placement of the stent, radiologically demonstrated, requires repeating the endoscopy and repositioning or replacement of the stent or the insertion of another stent (Figure 1 and Figure 2).

## 8. Contraindications

As important as determining when and which patients can benefit from stenting is identifying patients who would not benefit from stenting. Smith et al. in a recent study included among the contraindications for stent placement: complete, 360-degree circumferential interruption of the anastomosis and/or the presence of tracheoesophageal or bronchoesophageal fistula [15]. Additionally, patients with gastric tube ischemia and large anastomotic fistulas will not benefit from stenting, with other types of vacuum therapy procedures or surgical reintervention being indicated [17]. Other authors consider it necessary to drain a perianastomotic collection before mounting an esophageal stent [33].

## 9. Complications of Esophageal Stenting

Although we have noted that the method has a high-efficiency index in the management of postoperative fistula, it is important to know that stenting presents a series of complications, which determine specific morbidity and sometimes even mortality. Most studies note that the main complications related to the presence of the stent in the esophagus are tissue hyperplasia, mucosal erosion and parietal perforation, but endoscopic complications, associated with stent placement and removal, are also described; the most common complication is stent migration [37,38,57].

The rate of complications reported in the studies is variable, but one should be aware that these studies usually have a small number of patients included, and concerning the reporting of complications, some include all complications while others only the severe ones related to stenting. Aryaie et al. in a study on 20 patients showed an incidence of complications after esophageal stenting of 60%, including stent migration, mucosal friability, tissue integration and bleeding and aortoesophageal fistula (10%), leading to 1/20 death [24]. Freeman et al. reported stent-related morbidity in five patients (33%), including fistula persistence secondary to stent migration in 3/5 and esotracheal/bronchial fistulas in 2/5 [36]. Bohle et al. showed that 15% of the 63 patients with esophageal stent presented severe complications: parietal perforations at the distal end of the stent and erosion of the thoracic aorta or hepatic artery. Perforations were successfully treated by over-stenting or external drainage, and mortality was 1/63 due to mediastinitis. Arterial erosion was treated with intraluminal stenting of the aorta or hepatic artery. He noted that in 4 of the 5 patients with serious complications, a stent with a body diameter of 28 mm was used [16]. Van Boeckel et al. in a study in 2012 study reported a 10% rate of severe stent-related complications such as esophageal rupture, hemorrhage, stent migration with intestinal obstruction or death [39].

### 9.1. Stent Migration

Studies in the literature have reported stent migration in 10–40% of patients [20,24,25,35,40]. Stent migration rates vary depending on the diameter of the stent, the type of coating, the insertion site and the material used to make the stent [58]. Stenting a cervical anastomosis is a challenge from a technical point of view since it is usually located 2–4 cm distal to the cricopharyngeal muscle, so space is insufficient for anchoring the stent, the general indication being at least 5 cm. This is probably the main factor involved in stent migration identified by some authors [15], who report a migration rate of 62%, much higher than the rate of 8.5–47% reported for other locations [11,41]. The same authors suggested that delaying oral feeding may decrease migration rates [11]. Stent migration is due to discrepancies between the size of the esophagogastric lumen and the stent. However, no associations of the stent migration rate with the diameter of the stent body used, 20, 24 or 28 mm, were observed [16]. In a small pilot study, Fischer et al. [42] used a specially designed stent with a body diameter of 36 mm, partially covered, to prevent stent migration, but stent migration occurred in 4/11 patients. The rate of migration was significantly higher in procedures involving fully covered stents compared with partially covered and uncovered stents (*p* < 0.001) [21,41]. Although tissue growth over the exposed areas of the partially covered stent makes its removal more difficult [43] and may injure the mucosa, it serves as a natural form of stent fixation, as some authors have reported a lower risk of migration [43]. As a rule, in patients with stent migration, it is necessary to repeat the endoscopy, reposition the stent in the optimal position [15] or remove the migrated stent and insert a new stent in the correct position. Once this complication was resolved, it was observed that the fistula closed in all patients whose stents migrated [28]. Methods to minimize stent migration include selecting a larger stent diameter size (up to 28 mm), placing a longer stent and/or using endoscopic suturing of the proximal end of the stent to the esophageal mucosa. [11]. The choice of a larger stent size, however, should be weighed against the associated risk of erosion. A longer stent can minimize migration because the distal end of the stent can be used as an anchor in the gastric antrum [15]. Fixation in a certain position is a frequently used technique in medicine. Similar techniques have been used with stents in an attempt to reduce migration rates, including the use of metal endoscopic clips to attach the stent to the mucosa and transnasal anchoring of the proximal end of the stent [58]. Thus, some studies have shown that fixation significantly reduces migration. The studies of Rieder et al. and Wilcox et al. involving porcine models showed that endoscopic sutures increased the force required to dislodge the inserted stent [54,59], and clinical studies conducted by Sharaiha et al. and Fujii et al. showed a reduction in migration due to fixation [44,45]. In 2018, Law et al. presented the results of a meta-analysis that included 212 patients from 14 studies when endoscopic suture fixation was used, in which a migration rate of 15.9% was observed, this complication occurring in 1 out of 6 patients, despite excellent immediate technical success [46]. Jena et al. published a meta-analysis in 2023 that included nine studies on the anchoring of stents with the help of OTSC clips (conventional or Stentfix), demonstrating a lower risk of migration in the case of fixed stents compared to unfixed ones (RR = 0.24 (95%CI, 0.13–0.43, I^2^ = 0)) [47].

### 9.2. Tissue Overgrowth

Clinically significant tissue overgrowth was defined as stent-induced hypertrophic granulation tissue that either created dysphagia or made stent removal difficult [14]. In addition, tissue growth within and over the stent can lead to certain complications over time, such as stenosis and bolus impaction. Van Boeckel et al. identified tissue growth as a cause of complications in 15% of patients [39], while Kim et al. observed mucosal friability and mucosal stent integration in 20% of patients [28]. Of note, a stent should not be left for more than a month to avoid the growth of tissue, which may cause problems with its removal or erosion in adjacent structures. Rather, stent replacement is recommended if the patient continues to require this method to seal the fistula. Some authors have abandoned the use of partially covered stents [43]. This complication is described especially after the use of partially covered stents (Figure 3).

### 9.3. Erosion of the Mucosa

Some authors presented this complication in 2/63 patients, manifested by digestive hemorrhage, due to wall erosion by the distal end of the stent, which was successfully treated with interventional endoscopy [16]. A study by Speer et al. that investigated the risk factors of stenting complications found that preoperative radiochemotherapy was a significant risk factor for the occurrence of erosion (22.5 vs. 4.3%, *p* = 0.05). This study also noted a longer duration of stent maintenance in patients with major erosions compared to those without this complication, the difference being without statistical significance (92 vs. 36 days, *p* = 0.14) [14]. We encountered bleeding caused by mucosal damage in a patient with a stent placed for an anastomotic fistula. The hemorrhage occurred 3 weeks after the installation and was self-limited once the stent was removed (Figure 4).

### 9.4. Perforation in Nearby Organs

Speer et al. in 2016 reported major erosions with tracheoesophageal fistulas in 2/23 patients [14] and Al-Issa et al. in 2/15 patients [37], and Licht et al. described stent erosion in the pulmonary artery occurring in 1/31 patients [48], all requiring major reinterventions. In the literature, according to studies, it has been shown that most aortic fistulas appear after prolonged maintenance of the stent, from 26 to 36 days [49]. Haruštiak et al. described the presence of esoaerial fistula in 4/39 (10%) of stented patients [26]. We discovered the esotracheal fistula and proliferation of cartilaginous tissue, probably of tracheal origin in a patient after the extraction of the second stent, placed for a persistent anastomotic fistula 6 weeks after the insertion of the first stent (Figure 5 and Figure 6).

The mortality in a study by Lai et al. involved a patient who developed an aortoesophageal fistula secondary to stent placement. Aortoesophageal and aortoenteric fistulas are complications where major bleeding is often preceded by minor bleeding, which can be a warning sign [60,61,62]. Radial pressure from the stent can erode through the esophageal wall or induce inflammation that can lead to a fistulous tract between the aorta and esophagus. One of the two patients whose treatment was complicated by the formation of an aortoesophageal fistula first developed small bleeding before the massive hemorrhage that revealed the fistula. Aortoesophageal fistula is a fatal complication of stenting. An adequate analysis is necessary to assess the imminence of an aortoesophageal fistula with emergency vascular management in this case [14].

## 10. Perspectives

Of course, for a good period, esophageal stents represented an important step in the treatment of fistulas after esophagectomy in terms of solving this quite frequently encountered complication. However, along with the high-efficiency index in solving anastomotic fistulas, the method also records rates of failure or complications that can sometimes lead to death. The development of vacuum therapy has led, in the last period, to comparable or even better results in the efficiency of closing a fistula; therefore, more surgeons have favored the use of this therapy. By combining the benefits of the two methods, VACStent, an endoscopic stent associated with drainage capacity similar to vacuum therapy, is currently being put into practice. VACStent represents a hybrid method that combines the efficiency of esophageal stents completely covered with vacuum polyurethane foam attached to a continuous suction pump. One of the benefits is the prevention of stent migration by the suction force of the sponge cylinder immobilizing the VACStent on the esophageal wall, while at the same time, the attached external vacuum pump aspirates secretions and improves healing. Current studies on the results of this procedure are few as the procedure is in its infancy. In the first study reported in 2021, closure of the fistula was achieved after the failure of other methods [50].

The indications for VACStent installation are the following: spontaneous esophageal perforation, esophageal fistula and esophageal anastomosis fistula, occurring after surgical intervention [51]. According to a study by Lange et al. that included 15 patients, the healing rate of anastomotic fistulas after VACStent installation was 80%, without the occurrence of severe complications such as bleeding at the end of the stent or perforation, and stent migration was the most common complication [52]. This promising method can be applied in the centers that approve this procedure, and this method may take over some of the indications currently held by the two major endoscopic methods already established in the treatment of anastomotic fistulas.

## 11. Conclusions

Anastomotic leakage after esophagectomy will continue to be one of the most feared and costly postoperative complications. Given the significant impact on mortality and health care costs, surgeons are continually evaluating and improving their techniques to decrease the fistula rate. Management of such leaks require considerable expertise and judgment on the part of the surgeon; extensive knowledge of these treatment modalities helps promote a successful outcome.

Today, endoluminal stents are widely used in the management of anastomotic fistulas because they ensure the successful closure of defects while allowing for continuous enteral nutrition during recovery. Currently, the indications for stenting are defined for recent fistulas, smaller than 1/3 of the circumference of the anastomosis, without extralumenal cavities or necrosis of the adjacent tissues in a stable, non-septic patient. Severe stent-related complications appear to be preferentially associated with the use of large stents. Considering the correlation between the occurrences of perforations with the long duration of stent maintenance, it is recommended not to leave stents for more than 3–4 weeks. Studies with VACStent support the applicability of the method, but of course, studies on larger groups of patients are needed to demonstrate its place among the already established endoscopic methods. Although other procedures give promising results, this method has a well-defined role in the treatment of postoperative esophageal fistulas, but it is necessary to adapt the indication of the type of procedure for each individual patient.

## Figures and Tables

**Figure 1 life-13-00966-f001:**
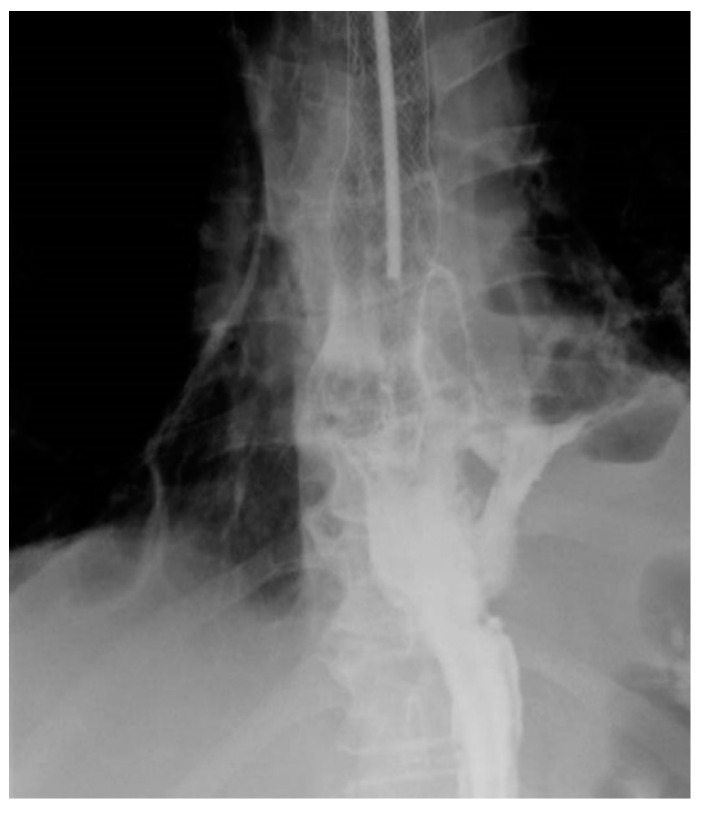
Persistence of the fistula after stent placement—Type 2, due to reflux of the substance at the level of the distal end.

**Figure 2 life-13-00966-f002:**
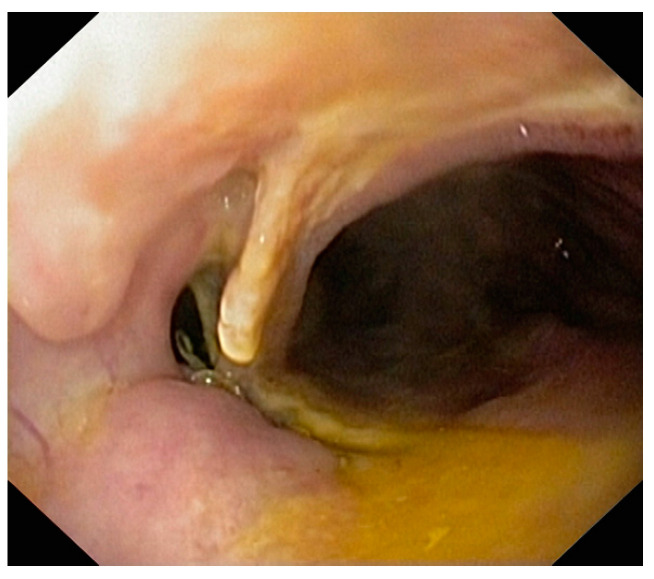
Persistence of the fistula of esogastric anastomosis after stent extraction at 3 weeks.

**Figure 3 life-13-00966-f003:**
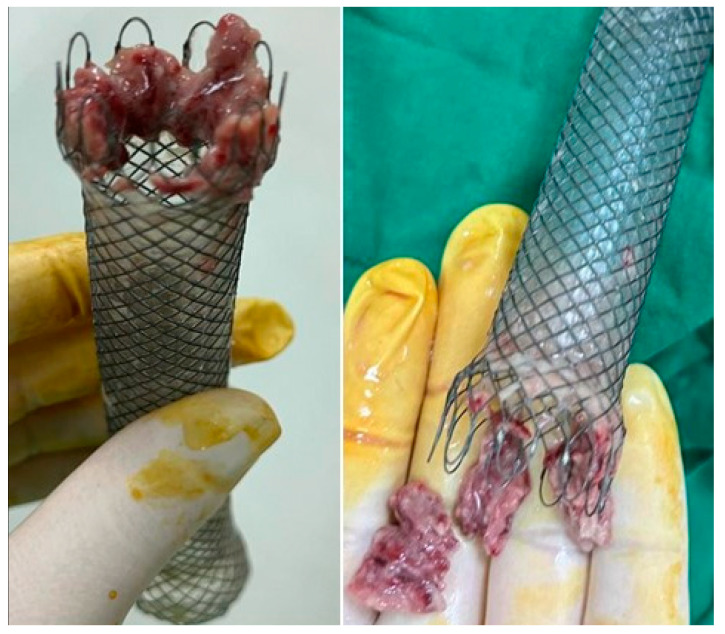
Impactation of the uncovered part with granulation tissue in a partially covered stent after maintaining it for more than 6 weeks.

**Figure 4 life-13-00966-f004:**
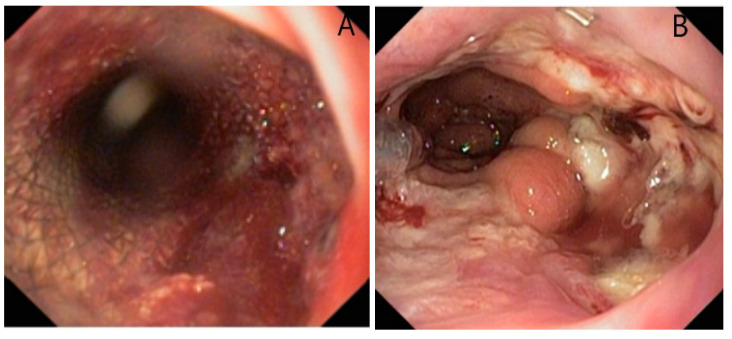
Hemorrhage due to mucosal erosion after 3 weeks (**A**), visible mucosal lesions after stent extraction; (**B**) esophagogastric anastomosis after Ivor Lewis procedure.

**Figure 5 life-13-00966-f005:**
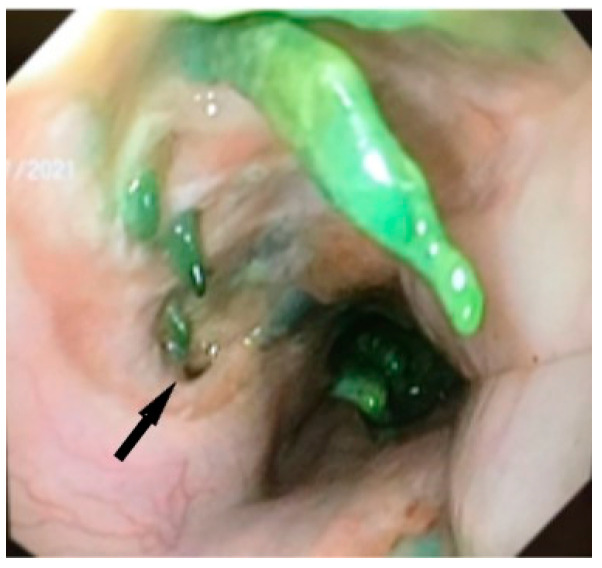
Esotracheal fistula (arrow) and the presence of cartilaginous formations in the esophagus after extraction of second stent at 6 weeks for persistent anastomotic fistula of esophagogastric anastomosis after Ivor Lewis procedure.

**Figure 6 life-13-00966-f006:**
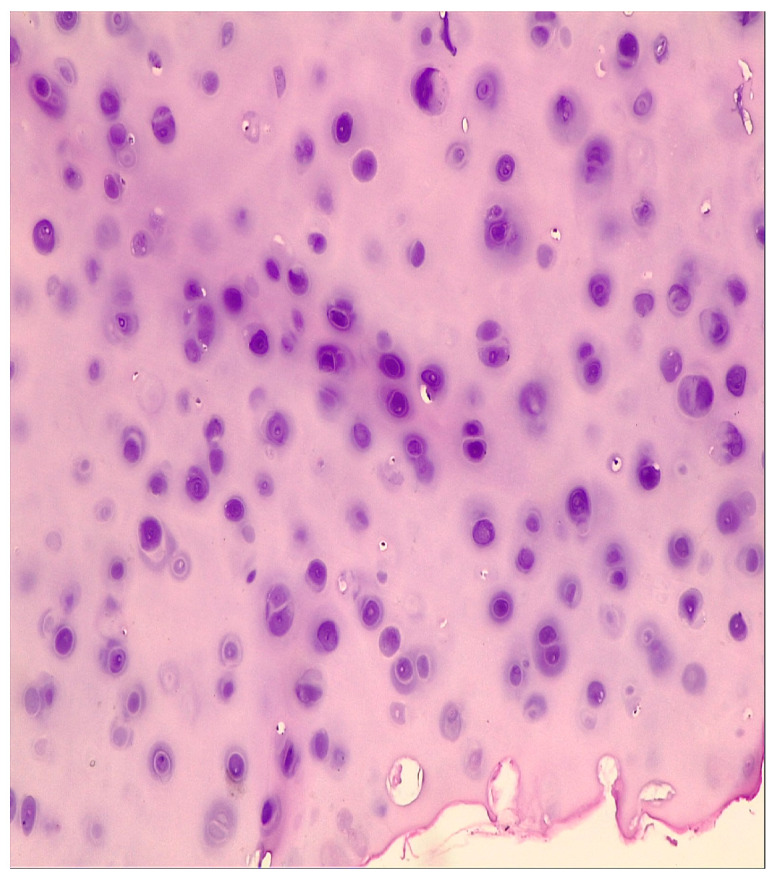
Histological structure of chondroid type, HE 10× (with the support of Dr. Enache Simona).

**Table 3 life-13-00966-t003:** Included studies that provided measurable results.

Ref.	Authors	Year	Type	PF (n)	Site	Stenting (n)	Stents (n)	Stent Healing	Stent Complications	Mortality
[4]	Schweeigert et al.	2014	OS	49		29/49		60%	10.3% (aortic erosion *n* = 3)	24.1%
[8]	Schaheen et al.	2014	R					72%		
[9]	Wu et al.	2017	OS	27	C			88.9%		
[10]	Lee et al.	2013	OS	20	A	20/20		95%		
[11]	Rajan et al.	2014	OS	32	T	32/32		96%	8.54%	
[12]	Schweigert et al.	2013	OS	38		22/38		77%	13.6% (aortic erosion *n* = 3)	22.7%
[13]	Nguyen et al.	2011	CS	18		9/28		100%		0%
[14]	Speer et al.	2016	OS	23	C	22/23	63/22	70%	62% migration, 11% tissue overgrowth, 8% minor erosion, 8% major erosion	0%
[15]	Smith et al.	2020	OS	11		10/11		90%		9%
[16]	Bohle et al.	2020	OS	34		34/34		76%	15%	20%
[17]	El-Sourani et al.	2022	OS	28		7/28		85.7%	28.6% (stent migration *n* = 1, perforation, *n* = 1)	0%
[18]	D’Cunha et al.	2011	OS	37		37/37	94/37	59%	8.1% (stent erosion, leak enlargement, fatal gastroaortic fistula)	0%
[19]	Stephens et al.	2014	OS	89		89/89		74.2%		3.4%
[20]	van Boeckel et al.	2011	SR	267		267/267		85%	34% (stent migration 31%, tissue overgrowth 3%)	
[21]	Singer et al.	2017	OS	21		21/21			19% (stent migration *n* = 4)	
[22]	Kucukay et al.	2013	OS	14		14/14		79%	21.4% (recurrent fistula *n* = 2 stent dislocation *n* = 1)	0%
[23]	Feith et al.	2011	SR	115		115/115		70%	Stent dislocation 53% after esophagocolonostomy, 61% with esophagojejunostomy, 49% with esophagogastrostomy.	
[24]	Aryaie et al.	2017	OS	20		20/20		90%	60% (stent migration *n* = 8, mucosal friability *n* = 4, tissue integration *n* = 2, bleeding *n* = 2, aortoesophageal fistula *n* = 2).	5%
[25]	Sanz Segura et al.	2022	OS	31		31/31		75%	21.7% (stent migration *n* = 5),	0%
[26]	Haruštiak et al.	2020	OS	80		39/80		62%	10% (airway fistula *n* = 4)	10%
[27]	Eizaguirre et al.	2016	OS	24		13/24		92.3%		7.7%
[28]	Kim et al.	2013	OS	33		4/33		50%		0%
[29]	Berlth et al.	2018	CS	111		76/111		73.9%	14.5% (perforation *n* = 1, gastrotracheal fistula *n* = 4, stenosis *n* = 5)	
[30]	Mennigen et al.	2015	CS	45		30/45		63.3%	36.7%	26.7%
[31]	Rausa et al.	2018	MA	134		80/134		pooled odds ratio of EVT/stent = 5.51	pooled odds ratio of EVT/stent = 0.38	pooled odds ratio of EVT/stent = 0.33
[32]	Tavares et al.	2021	MA	559				65.6%		
[33]	Hallit et al.	2021	MS	68		30/68		77%		6.7%
[34]	Hoeppner et al.	2014	OS	35		35/35	48/35	69%	71% (stent dislocation −19%)	0%
[35]	Freeman et al.	2012	CS	46		46/46		67.4%		
[36]	Freeman et al.	2011	OS	17		17/17		100%	18% (stent migration *n* = 3)	0%
[37]	Al-Issa et al.	2014	OS	20		15/20		67%	33% (stent migration *n* = 3, tracheoesophageal fistula, *n* = 2)	6.7%
[38]	Southwell et al.	2016	OS	21		21/21		95%	19 % (stent migration *n* = 4)	0%
[39]	van Boeckel et al.	2012	OS	32		32/32	83/52	76%	46% (tissue overgrowth *n* = 8, stent migration *n* = 10, ruptured stent cover *n* = 6, food obstruction *n* = 3, severe pain *n* = 2, esophageal rupture *n* = 2, hemorrhage *n* = 2)	2%
[40]	Iglesias Jorquera et al.	2021	OS	25		25/25	34/25	84%	28% (stent migration *n* = 7)	0%
[41]	Puig C et al.	2014	OS	21	A	21/21		19%	47% (stent migration *n* = 10)	
[42]	Fischer et al.	2013	OS	11		11/11		100%	36% (stent dislocation *n* = 4)	0%
[43]	Wei et al.	2013	OS	8		8/8	14/8		9/13 (tissue ingrowth *n* = 2, stent migration *n* = 10, esophageal lesion *n* = 1)	0%
[44]	Sharaiha et al.	2015	OS	21		37/37			37% stent migration	0%
[45]	Fujii et al.	2013	OS	18		18/18		56%	33% (stent migration *n* = 7, tracheoesophageal fistula *n* = 1)	5.6%
[46]	Law et al.	2018	MA	75				96.7%	15.9% stent migration	
[47]	Jena et al.	2023	MA					pooled clinical success rate of OTSC fixation = 0.79	pooled rate of migration following OTSC anchorage = 0.08	
[48]	Licht et al.	2016	OS	49		31/49		88%	3.2% (pulmonary artery erosion *n* = 1)	
[49]	Schweigert et al.	2011	OS	25		17/25		76.5%	23.5% (early recurrence *n* = 1, thoracic aorta erosion *n* = 3)	17.6%
[50]	Lange et al.	2021	OS	3		3/3		100%		
[51]	Chon et al.	2021	OS	10		15/10		70%		0%
[52]	Lange et al.	2023	MS	15		41/15				

PF—postoperative fistula, CS—comparative study, MA—meta-analysis, MS—multicentric study, OS—observational study, R—review, SR—systematic review, C—cervical, T—thorax, A—abdominal, EVT—endovacuum therapy, Ref.—reference, n—number.

## Data Availability

Not applicable.

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
