# Peer review of "The Use of Esophageal Stents in the Management of Postoperative Fistulas—Current Status, Clinical Outcomes and Perspectives—Review"

_life, 2023, doi:10.3390/life13040966_

Round 1
Reviewer 1 Report
The use of esophageal stents in the management of postoperative fistulas – current status and perspectives by Rodica Birla , Cristian Gelu Rosianu , Petre Hoara , Florin Achim , Alexandra Bolocan , Ahmed Mohssen , Narcis Copca , Silviu Constantinoiu is a review paper aimed to assess the updates on the use of endoscopic stents, for the treatment of the postoperative esophageal leak, in terms of indications, types of stents used, efficiency, specific complications and the perspectives.
The Authors reported that the endoscopic discovery of the fistula is usually followed by the insertion of a fully covered esophageal stent. It has an efficiency of more than 65% in closing the fistula, the failure is related to the delayed application of the method, a situation more suitable for endo vac therapy. The most common complication is migration, but life-threatening complications have also been described. The combination of the advantages of endoscopic stents and vacuum therapy is probably found in the emerging VACstent procedure. The Authors concluded that although the competing approaches give promising results, this method has a well-defined place in the treatment of esophageal fistulas. It is probably necessary to refine the indications for each procedure.
I think that the literature search was performed with a rigorous methodology.
However in my opinion the quality of the paper could be improved.
I would advise the Authors never to use terms such as "Another study" or "Another meta-analysis", but rather to indicate "The study by...." or "the meta-analysis by ......"
In addition, I would suggest making a table summarizing the 62 literature studies with the results of each paper, review or meta-analysis.
Finally, I consider it essential to insert the bibliographic reference in each statement of the review
Author Response
Reviewer 1 Thank you for your opinions and appreciations.
The use of esophageal stents in the management of postoperative fistulas – current status and perspectives by Rodica Birla , Cristian Gelu Rosianu , Petre Hoara , Florin Achim , Alexandra Bolocan , Ahmed Mohssen , Narcis Copca , Silviu Constantinoiu is a review paper aimed to assess the updates on the use of endoscopic stents, for the treatment of the postoperative esophageal leak, in terms of indications, types of stents used, efficiency, specific complications and the perspectives.
The Authors reported that the endoscopic discovery of the fistula is usually followed by the insertion of a fully covered esophageal stent. It has an efficiency of more than 65% in closing the fistula, the failure is related to the delayed application of the method, a situation more suitable for endo vac therapy. The most common complication is migration, but life-threatening complications have also been described. The combination of the advantages of endoscopic stents and vacuum therapy is probably found in the emerging VACstent procedure. The Authors concluded that although the competing approaches give promising results, this method has a well-defined place in the treatment of esophageal fistulas. It is probably necessary to refine the indications for each procedure.
I think that the literature search was performed with a rigorous methodology.
However in my opinion the quality of the paper could be improved.
I would advise the Authors never to use terms such as "Another study" or "Another meta-analysis", but rather to indicate "The study by...." or "the meta-analysis by ......"
- We corrected these expression in the manuscript
In addition, I would suggest making a table summarizing the 62 literature studies with the results of each paper, review or meta-analysis.
- We made table 3 with the included studies that provided measurable results
Finally, I consider it essential to insert the bibliographic reference in each statement of the review.
- We corrected the manuscript taking into account this opinion.
Reviewer 2 Report
This article is an interesting review about the management of postoperative fistulas and use of esophageal stents. The manuscript is well-written and readable; it contributes significantly to this field, collecting the most appropriate and recent references. The methodology is correct, included studies are well selected and selection process is accurate.
Although the management of this complication is still somewhat controversial, this review ensures that all indications and possible complications of placing an oesophageal stent are clearly set out. I have no comment for the authors. I only recommend a minor revision of English.
Author Response
Reviewer 2 Thank you for your opinions and appreciations.
This article is an interesting review about the management of postoperative fistulas and use of esophageal stents. The manuscript is well-written and readable; it contributes significantly to this field, collecting the most appropriate and recent references. The methodology is correct, included studies are well selected and selection process is accurate.
Although the management of this complication is still somewhat controversial, this review ensures that all indications and possible complications of placing an oesophageal stent are clearly set out. I have no comment for the authors. I only recommend a minor revision of English.
- We reviewed and corrected the manuscript.
Reviewer 3 Report
Interesting and excellent review. I recommend its publication, after reviewing the style.Author Response
Reviewer 3 Thank you for your opinions and appreciations.
Interesting and excellent review. I recommend its publication, after reviewing the style.
- We reviewed and corrected the manuscript.
Reviewer 4 Report
I have carefully read the text provided and I thank you for the opportunity to review it. The topic is certainly very interesting but the originality is low. There are already many examples of papers of this kind in the literature. The sections describing the use of esophageal stents and its management aspects in general are detailed and well done. However, in my opinion, the title should be reformulated since if the term "clinical outcomes" is included, something is expected also about therapeutic approaches. In this regard, I suggest including a quick review of the literature on the latest techniques for mitigating postoperative esophageal fistula in the discussion section, and I would like to recommend this paper: doi.org/10.1155/2020/8250904. As to make the text more interesting I would suggest to add someting about future prospects and robotic surgery in the discussion section. You could refer to this new paper : doi: 10.1007/s11605-023-05616-w
The written English needs to be revised as it is difficult to read. I recommend careful editing of the text.
Author Response
Reviewer 4 Thank you for your opinions and appreciations.
I have carefully read the text provided and I thank you for the opportunity to review it. The topic is certainly very interesting but the originality is low. There are already many examples of papers of this kind in the literature. The sections describing the use of esophageal stents and its management aspects in general are detailed and well done.
However, in my opinion, the title should be reformulated since if the term "clinical outcomes" is included, something is expected also about therapeutic approaches.
- We reformulated the title taking this observation into account.
“The use of esophageal stents in the management of postoperative fistulas – current status, clinical outcomes and perspectives – review”
In this regard, I suggest including a quick review of the literature on the latest techniques for mitigating postoperative esophageal fistula in the discussion section, and I would like to recommend this paper: doi.org/10.1155/2020/8250904.
- We added this paragraph: Lazzarin G studied 5 patients with eso-gastric fistula, after bariatric surgery, treated with double pigtail, and showed 100% healing, after 30-40 days. [37]
As to make the text more interesting I would suggest to add someting about future prospects and robotic surgery in the discussion section. You could refer to this new paper : doi: 10.1007/s11605-023-05616-w
- We added these references.
The written English needs to be revised as it is difficult to read. I recommend careful editing of the text.
- We reviewed and corrected.
Round 2
Reviewer 1 Report
I think the quality of the review paper has improved and therefore it can be accepted for publication.
Reviewer 4 Report
Thanks for corrections. In my opinion th papaer can be accepeted for pubblication in the present form